# Comprehensive Metabolite Profiling in Genetic Resources of Garlic (*Allium sativum* L.) Collected from Different Geographical Regions

**DOI:** 10.3390/molecules26051415

**Published:** 2021-03-05

**Authors:** Mostafa Abdelrahman, Sho Hirata, Takuya Mukae, Tomohiro Yamada, Yuji Sawada, Magdi El-Syaed, Yutaka Yamada, Muneo Sato, Masami Yokota Hirai, Masayoshi Shigyo

**Affiliations:** 1Botany Department, Faculty of Science, Aswan University, Aswan 81528, Egypt; meettoo2000@yahoo.com; 2Laboratory of Agroecology, Faculty of Agriculture, Kyushu University, Kasuya, Fukuoka 811-2307, Japan; hirata.sho.481@m.kyushu-u.ac.jp; 3Laboratory of Vegetable Crop Science, Graduate School of Sciences and Technology for Innovation, College of Agriculture, Yamaguchi University Yamaguchi City, Yamaguchi 753-8515, Japan; a003wfu@yamaguchi-u.ac.jp; 4Allium Unit, Division of Vegetable Breeding, Institute of Vegetable and Floriculture Science, National Agriculture and Food Research Organization (NARO), 360 Kusawa, Ano, Tsu, Mie 514-2392, Japan; yamatomo@affrc.go.jp; 5RIKEN Center for Sustainable Resource Science, 1-7-22 Suehiro-cho, Tsurumi-ku, Yokohama, Kanagawa 230-0045, Japan; yuji.sawada@riken.jp (Y.S.); yutaka.yamada@riken.jp (Y.Y.); muneo.sato@riken.jp (M.S.); masami.hirai@riken.jp (M.Y.H.); 6Molecular Biotechnology Program, Field of Advanced Basic Sciences, Galala University, New Galala City 43511, Egypt; magdiel_sayed@gu.edu.eg

**Keywords:** garlic, saponins, fructan, metabolome profiles

## Abstract

Garlic (*Allium sativum*) is the second most important *Allium* crop that has been used as a vegetable and condiment from ancient times due to its characteristic flavor and taste. Although garlic is a sterile plant that reproduces vegetatively through cloves, garlic shows high biodiversity, as well as phenotypic plasticity and environmental adaptation capacity. To determine the possible mechanism underlying this phenomenon and to provide new genetic materials for the development of a novel garlic cultivar with useful agronomic traits, the metabolic profiles in the leaf tissue of 30 garlic accessions collected from different geographical regions, with a special focus on the Asian region, were investigated using LC/MS. In addition, the total saponin and fructan contents in the roots and cloves of the investigated garlic accessions were also evaluated. Total saponin and fructan contents did not separate the garlic accessions based on their geographical origin, implying that saponin and fructan contents were clone-specific and agroclimatic changes have affected the quantitative and qualitative levels of saponins in garlic over a long history of cultivation. Principal component analysis (PCA) and dendrogram clustering of the LC/MS-based metabolite profiling showed two major clusters. Specifically, many Japanese and Central Asia accessions were grouped in cluster I and showed high accumulations of flavonol glucosides, alliin, and methiin. On the other hand, garlic accessions grouped in cluster II exhibited a high accumulation of anthocyanin glucosides and amino acids. Although most of the accessions were not separated based on country of origin, the Central Asia accessions were clustered in one group, implying that these accessions exhibited distinct metabolic profiles. The present study provides useful information that can be used for germplasm selection and the development of new garlic varieties with beneficial biotic and abiotic stress-adaptive traits.

## 1. Introduction

Garlic (*Allium sativum* L.) is one of the most economically important species of the genus *Allium* that has been used from ancient times as traditional medicine due to its nutritional, antimicrobial, and antioxidant properties, as well as a spice and herb to heighten the flavor of the dishes and health quotient [1,2,3]. The total world garlic production in 2018 was 28.5 million tonnes (MT), of which, 26.1 MT were produced in Asia, 0.86 in Europe, 0.83 MT in the Americas, and 0.73 MT in Africa [4]. Currently, numerous garlic products, including garlic powder, garlic paste, picked garlic, garlic vinegar, and garlic slices, have become available in the food markets to suit the growing demands of consumers [5]. However, the geographical origin of cultivars, cultivation methods, environmental conditions, and food processing have significant effects on the quality of garlic and garlic products, especially the contents of organosulfur compounds, amino acids, sugars, and phenolics [6,7,8]. For example, using a high temperature during thermal processing decreased amino acid, allicin, and moisture contents of black garlic, whereas increases in the total phenolic, reducing sugar, and organic acid contents were observed, indicating that temperature had a significant influence on the quality and flavor of black garlic [7].

The center of origin for garlic is considered to be the northwestern side of the Tien Shan Mountains in Central Asia due to the discovery of some fertile clones of a primitive garlic type in this area [9], leading to new avenues of research on garlic breeding, genetics, and physiology. Garlic has a large diploid genome (2*n* = 2*x* = 16), with an estimated haploid size of 15.9 gigabase pairs, which is ~32 times greater than the rice (*Oryza sativa*) genome [10]. Garlic is a usually sterile plant and does not produce fertile seeds by sexual reproduction, and thus, clonal propagation by its cloves is the main cultivation method [11,12]. Although the peculiar garlic reproduction could lead to low genetic diversity [13], garlic shows surprisingly high biodiversity, as well as phenotypic plasticity and environmental adaptation capacity [14,15]. For example, garlic genotypes vary considerably in their ability to produce fertile pollen, leaf number, floral scape length and inflorescences, and receptive stigmas [11,16]. The reason for this large biodiversity in garlic is not fully understood, which could be partially attributed to the large genome with many multicopy genes and duplications, including noncoding sequences and tandem repeats that have been accumulated through the history of domestication [17,18,19].

Garlic cultivars vary not only in their fertility but also in their chemical compositions, including organosulfur compounds, saponins, phenolic compounds, and polysaccharides [4,20,21]. In general, diallyl sulfide (DAS), diallyl thiosulfonate (allicin), diallyl trisulfide (DATS), diallyl disulfide (DADS), S-allyl-cysteine (SAC), and S-allyl-cysteine sulfoxide (alliin) are the main organosulfur compounds detected in garlic [22,23]. In addition, the total amount of saponin in purple garlic was almost 40 times greater than that in white garlic, and several saponin compounds were only found in purple garlic [24]. Moreover, β-resorcylic acid, followed by pyrogallol, gallic acid, rutin, protocatechuic acid, and quercetin, are the major phenolic compounds detected in different garlic cultivars [25]. The characterization of garlic germplasm has been largely based on phenotypic characteristics, but these can vary under various agroclimatic conditions [26]. Although garlic is produced in a wide geographic range, there have been few studies on the chemical compounds in garlic based on the geographical origin. Therefore, the objective of this study was to investigate the metabolite profiles of 30 garlic accessions collected from different geographical regions using LC/MS. In addition, the total saponin and fructan contents, as well as the morphological traits (plant height, number of leaves, leaf width, and sheath diameter), were also evaluated. This study provides in-depth insight into the metabolite richness in different garlic accessions based on the geographical distribution, which could be useful information that enables us to understand the phenotypic plasticity and environmental adaption of garlic germplasm.

## 2. Results

### 2.1. Quantitative Analysis of Saponin and Fructan Contents in the Examined Garlic Accessions

Saponin contents in garlic root tissues varied between 3.36 to 32.18 mg g^−1^ DW, and the average was 14.49 mg g^−1^ DW (Appendix A, In the Appendix A). Garlic accessions showed considerable variations in their saponin contents, regardless of their country of origin, indicating that the production of saponins in the examined garlic accessions was not influenced by the geographical distribution (Figure 1A, Appendix A). For example, five accessions, namely, EGY489, JPN_Taishu-san, SYR_No.5, JPN_Kikai-onodu, and SAN BARTOROME_Gatur, obtained from Egypt, Japan, Syria, Japan, and Turkey showed higher saponin contents (32.18, 30.41, 29.82, 24.96, and 23.68 mg g^−1^ DW, respectively) among the 30 investigated garlic accessions (Figure 1A, Appendix A (In the Appendix A)). All these five accessions originated from Egypt, Japan, Syria, and Turkey, respectively (Table 1). On the other hand, five accessions, namely, JPN39, TUR542, EGY55, F189, and IND-IIT exhibited lower saponin contents (3.36 to 7.12 mg g^−1^ DW), whereas 16 accessions (JPN65, CHN54, PER137, Fs407, GER556, Fs414, CHN180, Fs422, JPN40, TWN45, F138, F1-200-34, GRE307, VNM_Mai Dinh, JPN37, and F115) showed moderate levels ranging between 7.82 to 18.21 mg g^−1^ DW (Figure 1A; Appendix A (In the Appendix A)). The histogram distribution of the saponins indicated that most of the investigated garlic accessions exhibited moderate saponin contents (Figure 1A).

Likewise, the fructan contents showed quantitative variations (2.58 to 23.0 g inulin equivalent per 100 g FW) in the cloves of investigated garlic accessions, and the geographical distribution did not show any significant influence on the fructan profile (Figure 1B, Appendix A). For example, Fs407, followed by JPN40 and CHN524, showed higher fructan contents (23.0, 19.48, and 19.18 g inulin equivalent 100 g^−1^ FW), whereas F189, SHA_Shanghi, Fs414, IND-IIT, F1-200-34, VNM_Mai Dinh, and GER556 displayed lower fructan contents ranging between 2.58 to 10.93 g inulin equivalent 100 g^−1^ FW (Figure 1B, Appendix A). The remaining garlic accessions exhibited considerable amounts of fructan ranged between 11.08 to 18.91 g inulin equivalent 100 g^−1^ FW (Figure 1B, Appendix A). Although these garlic accessions were grown in similar agroclimatic zones at the field experiment of Yamaguchi University, the saponin and fructan contents were clone-specific and displayed considerable variability, indicating that these biochemical traits might be genetically controlled in garlic.

### 2.2. Metabolic Variations in the Leaf Tissues of the Investigated Garlic Accessions

In the present study, retrieval of the widely targeted metabolome profiles in the leaf tissues of 30 garlic accessions was carried out. A total of 472 metabolite intensities were obtained using LC/MS analysis. The metabolite data matrix was further filtered by removing the metabolites with a signal-to-noise ratio (S/N) < 3, leaving 101 metabolites for further analysis (Appendix A). All the 101 metabolites were subjected to multivariate principal component analysis (PCA) to reduce the dimensionality of the data set with many variables being correlated with each other while retaining the variation present in the dataset up to the maximum extent (Figure 2A). All the metabolite variables were loaded into two major principal components (PC1 and PC2), explaining 38.9% of the total variance (Figure 2A). PC1 explained 25% of the variance, whereas PC2 explained 13.9% (Figure 2A). The PCA plot based on a K-mean cluster divided the garlic accessions into two major clusters (Figure 2A). Cluster I contained Japanese (JPN37, JPN39, and JPN40), Central Asian (F115, F138 and F189, Fs407, Fs414, and Fs422), Taiwanese (TWN45), Chinese (CHN524), Vietnamese (VNM_Mai Dinh), Syrian (SYR_No.5), Peruvian (PER137), and Egyptian (EGY489) accessions, whereas cluster II contained Japanese (JPN_Kikai-onodu, JPN_Taishu-san, and JPN65), Chinese (SHA_Shanghi, CHN54, and CHN180), Central Asian (F1-200-34), Thai (THA67-4 and THA16-5), Indian (IND-IIT), Turkish (TUR542 and Gatur), and Egyptian (EGY55) accessions (Figure 2A).

Next, we identified the top metabolites that exhibited strong contributions to both PC1 and PC2 (Figure 2B). Allo-threonine (X00695), threonine (X200061), leucine (100024), saccharopine (X00502), and leucine_isoleucine (X100043) were among the top five metabolite variables that showed high contributions to PC1, whereas quercetin-3,4′-O-di-beta-glucopyranoside (X01219), alliin (X260007), isorhamnetin-3-O-glucoside (X00419), S-carboxymethyl-L-cysteine (X00338), and methiin (X260006) exhibited high contributions to PC2 (Figure 2B). Next, we conducted dendrogram clustering as a complementary to PCA, where the results showed a similar separation pattern as given by PCA, suggesting that the garlic accessions in the two clusters exhibited different metabolic profiles (Figure 2C). Although the metabolite profiles of the investigated garlic accessions were not separated based on the geographical distribution, most of the Central Asian accessions were clustered in one group, implying that garlic that originated from Central Asia might have similar metabolic profiles (Figure 2A–C). A Pearson correlation coefficient (Pcc) was carried out using a normalized metabolite matrix to identify the relationship among garlic accessions based on their metabolite level (Figure 3A). Our result indicated highly significant positive correlations between most of the garlic accessions, especially between the F138, F189, Fs 407, Fs414, JPN37, JPN39, TWN45, EGY489, and GER556 accessions (*r* > 0.95), whereas Fs422 showed negative or weak correlations with the other garlic accessions (Figure 3A).

Finally, heatmap clustering was carried out using a normalized metabolite data matrix (Figure 3B). The heatmap cluster separated the garlic accessions into two major clades similar to the PCA results, with some variations. For example, the JPN37, JPN39, F115, F138, Fs407, Fs414, VNM_Mai Dinh, TWN45, and EGY489 accessions, which were placed in cluster I, were characterized by high choline chloride, glutamic acid, luteolin-7-O-glucuronide, kaempferol-3-glucuronide, quercetin-3-glucuronide, and alliin (Figure 3B). On the other hand, amino acids, lysine, glutamine, and serine were highly accumulated in the SYR_No.5, JPN40, and SHA_Shanghi accessions in cluster II, whereas the GER307, JPN_Taishu-san, JPN_Kikai-onodu, and THA16-5 accessions showed higher accumulations of cyanidin-based and luteolin-based glucosides (Figure 3B). In general, the metabolome analysis suggested that the geographical distribution had limited effects on the garlic metabolome profile, and most of the variation could have been cultivar specific. In addition, several accessions that were derived from Central Asia showed similar metabolic profiles, which could be an interesting germplasm for further genetic studies to understand the metabolome plasticity in garlic under current climatic changes.

## 3. Discussion

The authentication of garlic based on geographic origin using metabolome profiling is currently a challenging research topic. However, the use of fertilizers, which may change the mineral profiles in soil or modify their availability as a crop due to changes in the pH, can significantly affect garlic metabolite contents [27]. Thus, in order to categorize garlic accessions, the cultivation of garlic needs to be carried out under the same agro-ecological conditions to minimize environmental effects [28]. With regard to all these factors, we decided to employ a metabolomics-based strategy, presuming that not only the garlic cultivar but also the growing conditions in each respective locality would influence each metabolome’s characteristic pattern. A total of 30 garlic accessions derived from 13 different countries, namely, Japan, Taiwan, China, Central Asia, Vietnam, Thailand, India, Syria, Egypt, Turkey, Germany, Greece, and Peru, were grown at an experimental field station for two successive seasons in an experimental field at the Institute of Vegetable and Floricultural Sciences, National Agriculture and Food Research Organization (NARO), Japan, under the same conditions. The roots and cloves were used for determining the total saponin and fructan contents, respectively, whereas the leaves were subjected to a targeted metabolome screening approach based on high-resolution mass spectrometry. The combination of both conventional and mass spectrometry analyses provided a broader range of information regarding each respective sample.

The average content of the total saponins in dry roots of the investigated garlic accessions was 14.49 mg g^−1^ DW, where the EGY489 accession showed the highest total saponin content in comparison with the other garlic accessions of 32.18 mg g^−1^ DW, followed by the JPN_Taishu-san and SYR_No.5 accessions with 30.41 and 29.82 mg g^−1^ DW, respectively (Figure 1A, Appendix A). On the other hand, JPN39 and TUR542 showed lowest saponin contents with 3.38 and 3.36 mg g^−1^ DW, respectively (Figure 1A, Appendix A). These results indicated that the saponin contents are highly variable among the investigated garlic accessions. Saponin is a well-known antifungal compound that protects plants against a wide range of phytopathogens, where our recent studies [29,30,31] indicated that the increase in saponin contents was positively correlated with improved disease resistance against *Fusarium* and *Phomopsis* pathogens in *Allium* and *Asparagus* crops, respectively. Although steroidal saponins are widely distributed in the genus *Allium* [31], there is limited information regarding the total saponin contents in garlic germplasm. For example, in *Allium nigrum*, a Middle Eastern species of wild onion known as black garlic, exhibited high total saponin contents in the roots (19.38 mg g^−1^ DW), whereas lower saponin contents were detected in the leaves (10.48 mg g^−1^ DW), indicating that there is a dynamic variance trend in total saponins between different organs [32]. Likewise, the total saponin contents in the cloves of purple and white garlic varieties were 0.36 and 14.10 mg, respectively, and the total number of saponins was almost 40-fold higher in purple garlic than in the white variety, especially in the external part of the cloves [24]. In addition, β-chlorogenin is the sapogenin of spirostanol saponin eruboside B, and β-chlorogenin can mainly be detected in garlic cultivars and in small amounts in elephant garlic (*Allium ampeloprasum*), whereas most of the *Allium* vegetables did not contain β-chlorogenin, implying that garlic has its specific saponin profiles and β-chlorogenin may be a suitable biomarker for garlic [33]. In this study, the saponin contents were not different between regions; however, they varied between accessions (Figure 1A, Appendix A), implying that each accession seemed to produce its own specific saponins. Thus, it is highly probable that the agroclimatic changes affected the quantitative and qualitative levels of saponins in garlic over a long history of cultivation as an adapted mechanism against pathogens in various agroclimatic regions. For example, the high saponin contents in the EGY489 and SYR_No.5 accessions might have been associated with the hot climatic condition in the Mediterranean region, which is favorable for pathogenic fungi; thus, the increase in saponin contents might have been an adaptive mechanism in these accessions against pathogens. The role of saponin in garlic’s defense against pathogens has been reported, for example, eight new saponin compounds were isolated from *Allium sativum* L. var. Voghiera and named voghieroside A1/A2, B1/B2, D1/D2, and E1/E2 based on the aglycon structure [34]. In addition, two known spirostanol saponins, namely, agigenin 3-O-trisaccharide and gitogenin 3-O-tetrasaccharide, were isolated from *A. sativum* L. var. Voghiera [34]. The isolated saponin compounds exhibited an effective antifungal activity against the non-pathogenic *Trichoderma harzianum* and the pathogenic *Botrytis cinerea*, indicating that saponin plays a pivotal role in garlic’s defense against fungi pathogens [34]. Thus, breeding programs aiming to increase the saponin contents in modern garlic cultivars might be an effective strategy to increase yields and improve garlic quality.

The average content of fructan in the cloves of all the investigated garlic accessions was 14.19 g inulin equivalent per 100 g FW, where the Central Asian accession Fs407 showed the highest fructan content in comparison with the other investigated garlic accessions with 23.0 g 100 g^−1^ FW, followed by the Japanese accession JPN40 with 19.48 g 100 g^−1^ FW (Figure 1B, Appendix A). On the other hand, F189 showed the lowest fructan content with 2.58 g 100 g^−1^ FW (Figure 1B, Appendix A). Carbohydrates are one of the major constituents of garlic cloves, where fructan, a high molecular weight polymer, is the primary storage carbohydrate in garlic cloves [35,36]. In garlic, the changes in fructan and sugar metabolisms were correlated with a higher growth rate and shortened crop cycle after clove replanting, and differential degradation or synthesis of fructan may result in the altered availability of simple sugars for the sink during garlic storage [37]. For example, insufficient cold conditions can significantly affect garlic plant development, causing a modification in the fructan profile and subsequently causing nondifferentiated bulbs, which have no commercial value and represent almost 50% of crop losses [38]. For example, the storage temperature has a pivotal influence on the chemical composition of garlic cloves, where the storage of garlic cloves at low temperature (5 °C) has been reported to affect the expression of the *sucrose:sucrose 1-fructosyltransferase* gene, which is associated with fructan metabolism and consequently with the carbohydrate and total soluble solids content [38]. Our results and previous examples indicated that the fructan content in garlic results from the interaction between the location and genotype [39]. In plants, fructan acts as a reserve carbohydrate and protects the plant against cold and drought stress during the dormancy period [40,41]. For example, a high level of fructans was observed in transgenic tobaccos with a drought-resistant gene, which contributed to the protection of the membrane and other cellular components by inducing cell wall hardening and limiting cell growth to reduce water demands [40]. Similarly, a high level of fructan content was observed in Indonesian shallots (*Allium cepa* aggregatum group) in comparison with Japanese bulb onions, which was linked with drought stress tolerance and protection against fungal and pathogenic attacks [42,43]. In the present study, the garlic accessions characterized by high fructan accumulation open the possibility of developing high-value-added garlic production in arid climate conditions.

Metabolite profiles of the 30 accessions of garlic were obtained by using LC/MS and combined with multivariate statistical analysis (Figure 2A). In the PCA score and loading plots, clear separations were shown between garlic accessions, where most of the Central Asian, Japanese, Taiwanese, and Vietnamese accessions were grouped together in cluster I (Figure 1A), whereas the Egyptian, Turkish, Greek, Indian, Chinese, and two Japanese accessions were grouped in cluster II (Figure 2A). The top 40 compounds with high contributions to PC1 and PC2 were identified (Figure 2B). Likewise, the dendrogram cluster separated the investigated garlic accessions into two groups similar to those found in the PCA results. PCA is a useful statistical tool for exploratory data analysis and making predictive models, which can be helpful with visualizing relatedness between populations and genetic distance [44]. Although the geographical location did not show clear effects on the metabolite levels of the 30 accessions, Central Asian accessions, including F115, F138, F189, Fs407, Fs414, and Fs422, were combined together in cluster I, indicating that the Central Asian accessions might have a distinct metabolite profile compared with other garlic accessions (Figure 2A–C). Among the characteristic metabolites, choline chloride, glutamic acid, luteolin-7-O-glucuronide, kaempferol-3-glucuronide, quercetin-3-glucuronide, alliin, and methiin were highly accumulated in Central Asian accessions, as well as other Japanese, Taiwanese, and Chinese accessions, which were grouped in cluster I (Figure 2A–C and Figure 3B). On the other hand, the garlic accessions that were grouped in cluster II, especially the Thai (THA16-5 and THA67-4), Japanese (JPN40, JPN65, JPN_Taishu-san, and JPN_Kikai-onodu), Indian (IND-II), and Chinese (SHA-Shanghi, CHN65) accessions, were characterized by high-anthocyanin-related metabolites, including cyanidin-based and luteolin-based glucosides, as well as amino-acid-related metabolites (Figure 2A–C and Figure 3B). The high-anthocyanin-related metabolites in the garlic accessions of cluster II might have contributed to the abiotic and biotic stress tolerance in these accessions. On the other hand, the high alliin and methiin contents in the Central Asian accessions might be useful agronomic traits for improving the taste and aroma in garlic. It is worth noting that the addition of sulfur fertilizer improved the alliin contents in garlic and onion plants, implying that garlic cultivars with a high affinity toward sulfur uptake might be a candidate germplasm for increasing organosulfure compounds and the subsequently taste and aroma [45]. *Allium* species are rich in flavonol and anthocyanin glucosides, which play important roles in plant protection against UV stress and phytopathogens [31,46,47]. Thus, the identified garlic accessions with high flavonoid contents might be useful genetic resources for garlic breeding and genetic programs for the development of novel garlic varieties with higher flavonoid contents for cultivation in harsh environmental conditions. In our previous studies, we investigated the relationship between the geographical distribution of these garlic accessions and the major biomorphological characteristics and discussed the various characteristics of garlic and its high environmental adaptability [1,48]. In this present study, the obtained metabolome-profiling data in these accessions were not closely related to the geographical distribution. The results of this present study suggest that garlic contains accession-specific characteristics in addition to regional characteristics, which is consistent with the results of our previous studies [1,48]. Since the traditional landraces of garlic are under severe threat of extinction due to the rapid replacement of modern cultivars, which are easier to cultivate, it is necessary to maintain and preserve the local varieties in each region for the future utilization of garlic [49]. The accessions from Central Asia showed similar metabolome profiling and different tendencies to the other accessions. Since this region is around the origin of garlic and genetic diversity is likely to remain high [47,48], it is expected to have unique traits that are not found in existing modern cultivars [49]. In addition, some accessions also showed unique metabolite-profiling traits. In these accessions, more detailed studies on, for example, the level of genetic diversity and/or tolerance to environmental stresses are necessary in the future. Furthermore, the details of the spread of garlic to different parts of the world are often unknown or complicated [50]. The analysis and collection of huge amounts of data, such as many kinds of metabolome profiling data, as shown in this study, or next-generation sequencing data, could be an important tool for discussing the spread routes of garlic.

## 4. Materials and Methods

### 4.1. Plant Materials

Bulbs of 107 garlic accessions have been collected from around the world since the 1970s and were managed by Kagoshima University, Japan (31.56° N, 130.54° E), and bulbs of 33 garlic accessions were managed at Saga University, Japan (33.24° N, 130.29° E) until 2012, when the management of both collections was taken over by Yamaguchi University, Japan (34.14° N, 131.47° E). To maintain these important garlic genetic resources, ten cloves from three to five bulbs of each accession were vegetatively propagated every year in October and harvested by the end of June. Then, the entire garlic plant (bulb, roots, and stalk) were hung and kept in a dry place with good air ventilation inside the Greenhouse. Out of these harvested plants, 30 accessions were used in this study. The 30 garlic accessions contained 7 clones from Central Asia, 6 clones from Japan, 4 clones from China, two clones from Thailand, one clone from Taiwan, one clone from India, one clone from Syria, two clones from Egypt, two clones from Turkey, one clone from Germany, one clone from Greece, and one clone from Peru (Table 1). Detailed information regarding the accessions was reported [1,48,49,50,51]. These bulbs were stored at 4 °C under dark conditions in the summer. Then, a compound fertilizer was applied before planting and 100 kg/ha ammonium sulfate as the N source and to adjust the soil pH (pH 6.0–6.5), 100 kg/ha potassium chloride as the K source, and 120 kg/ha calcium superphosphate as the P source were the three major nutrients in the basal dressing fertilizer. The garlic collections were planted in an experimental field at Yamaguchi University in October. For each clone, eight cloves per accession were randomly selected and planted. The row and plant-to-plant spacing were 20 cm, and the depth of the seeding row was 10 cm. All garlic accessions were harvested at the end of June or the beginning of July. Under these growing conditions, the quantitative changes in the fructan and saponin contents in the developed cloves and roots of each garlic accession were investigated.

For the metabolome analysis, garlic accessions were cultivated in an experimental field at the Institute of Vegetable and Floricultural Sciences, NARO, Mie, Japan (34.77°N, 136.43°E). The garlic accessions were planted in the same way as at Yamaguchi University described above (the previous crop was plowed into the soil after growing Sorghum). CDU-S555 (JCAM AGRI.Co., Ltd., Tokyo, Japan), Ecolong 413-100 (JCAM AGRI.Co., Ltd., Tokyo, Japan), and Ecolong 413-140 (JCAM AGRI.Co., Ltd., Tokyo, Japan) were applied at 130, 360, and 130 kg/ha, respectively before planting. The field experiment was carried out for two successive seasons (2017–2018 and 2018–2019). Two months after planting, the youngest leaves, which were over 4 cm in length, were sampled. Then, the leaf tissues were immediately placed in 2 mL tubes, immersed in liquid nitrogen, and stored at −80°C for further analysis. The monthly mean temperature (°C), precipitation (mm), and accumulation of daylength (h) (minimum/maximum) were 16.4 °C (4.7 °C/29.2 °C), 191.6 mm (17.0 mm/648.0 mm), and 194.3 h (95.0 h/279.2 h) in 2017–2018, while they were 16.8°C (6.0 °C/28.6 °C), 108.6 mm (22.0 mm/342.5 mm), and 174.4 h (103.3 h/262.1 h) in 2018–2019, respectively (data from the Japan Meteorological Agency [52]).

### 4.2. Determination of the Total Saponins in the Garlic Accessions

The saponin contents were obtained from the root tissues of 26 garlic accessions during a doctoral study [48,51] (Appendix A). After the harvest, the developed roots in each accession were washed to remove soils. Then, all accessions were dried in a vented greenhouse for a month to obtain dry roots. The total saponin contents from the root portions of all accessions were extracted in accordance with the methods of [31,32]. Briefly, the dry roots of each accession were bulked together and exhaustively extracted at room temperature with the solvent n-hexane to remove nonpolar compounds. The defatted materials were extracted with 80% methanol for 30 min of sonication and filtrated twice. The extract was dried in a rotary evaporator with vacuum pump v-700 (Büchi^®^, Rotavapor^®^ R-3; Merck KGaA, Darmstadt, Germany) under reduced pressure at 50 °C and then partitioned between butanol (BuOH) and H_2_O (1:1). The BuOH layer was filtered and then concentrated under vacuum, causing a crude extraction of saponins. The total saponin content in the crude extract was determined spectrophotometrically at 473 nm using a 0.7% vanillin-60% H_2_SO_4_ reagent. The absorbance was measured three times against a blank at 473 nm using a U-2000 spectrophotometer (Hitachi High-Technologies Corporation, Tokyo, Japan). The major saponin compound in garlic, namely, β-chlorogenin [53], was used as a standard to establish a calibration curve. The total saponin compound content was expressed as the β-chlorogenin equivalent per gram dry weight root (mg g^−1^ DW root).

### 4.3. Determination of the Fructan Contents in the Garlic Cloves

The fructan contents were obtained from the cloves of 27 garlic accessions during a doctoral study [48,51] (Appendix A). The fructan content in the cloves was also determined using the thiobarbituric acid method [54] with minor modifications. To determine the fructan alone, sucrose was first removed via digestion with invertase. In addition, free fructose was removed from the extracts by heating an aliquot in 1N NaOH at 100 °C for 10 min. A 20 μL aliquot of the extract was incubated with 10 μL 2 mg/mL invertase (Invertase from baker’s yeast, Merck KGaA, Darmstadt, Germany) and 10 μL 25 mM ammonium acetate buffer (pH 5.5) for 5 min. The fructan contents of the garlic were expressed as the gram inulin equivalent per 100 g fresh weight (g 100 g^−1^ FW).

### 4.4. Targeted Metabolome Profiling of the Garlic Accessions

The leaf tissues were freeze-dried using a rotary evaporator (Büchi^®^, Rotavapor^®^ R-3;) coupled with a vacuum pump v-700 (Büchi^®^, Rotavapor^®^ R-3) under reduced pressure at 10 ± 1 °C. The sample preparation process was automatically performed using a liquid-handling system (Microlab Star Plus, Hamilton, Birmingham, United Kingdom), as previously described [55]. Briefly, 4 mg dry weight of leaf tissues was accurately weighted and transferred into a 2 mL tube with a 5 mm zirconia bead YTZ-5 (Watson, Co., Ltd., Murotani, Japan). The metabolites were extracted using a proportional volume of 4 mg mL^−1^ extraction solvent (80% methanol, 0.1% formic acid, 210 nmol L^−1^ 10-camphorsulfonic acid, and 8.4 nmol L^−1^ lidocaine as internal standards) using a multibead shocker (Shake Master NEO, Bio Medical Science, Tokyo, Japan) at 1000 rpm for 2 min. After the centrifugation, the extracts were diluted to 400 µg mL^−1^ in first season and 1000 µg mL^−1^ in the second season using an extraction solvent. Then, 250 µL of the extract was transferred to a 96-well plate, dried, redissolved in 500 and 250 µL of ultrapure water, in season one and season two, respectively, and filtered using MultiScreen_HTS_384 well (Merck KGaA, Darmstadt, Germany). One and 1.5 µL of the solution extract at a final concentration of 20 and 100 µg mL^−1^, in season one and season two, respectively, was subjected to widely targeted metabolomics using LC/MS (UPLC, Nexera MP, and LCMS-8050, Shimadzu, Japan) in the multiple reaction monitoring mode; for detailed information, see Appendix A.

### 4.5. Data Analysis

The final data results were obtained by taking the average of the results of two successive seasons. The histogram, PCA, heatmap, and correlation analyses were carried out using R v.4.0.4 (www.r-project.org (accessed on 2 January 2021)).

## Figures and Tables

**Figure 1 molecules-26-01415-f001:**
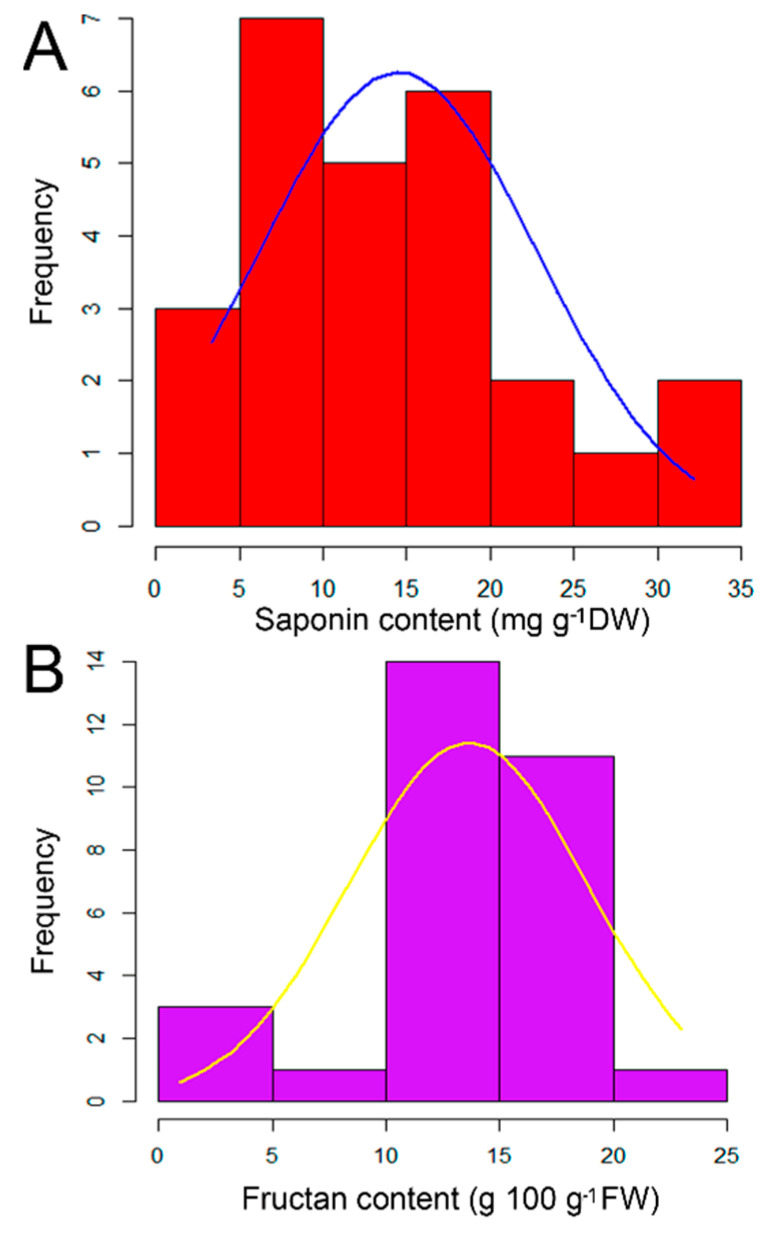
Histogram distribution of saponin (**A**) and fructan (**B**) contents in the investigated garlic accessions.

**Figure 2 molecules-26-01415-f002:**
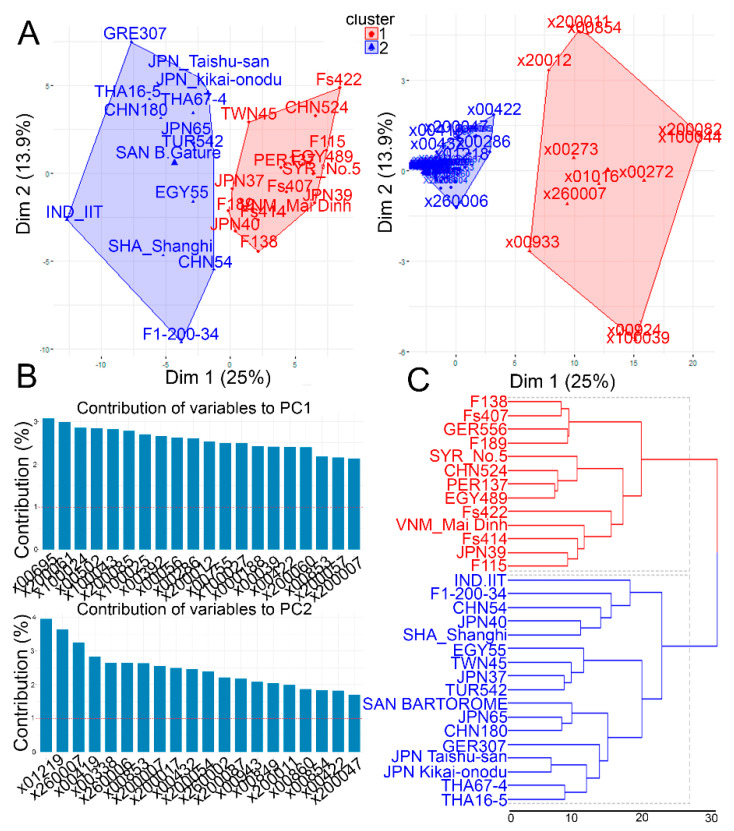
Principal component analysis (PCA) and dendrogram clustering of the normalized metabolite intensities in 30 garlic accessions: (**A**) PCA loading and score plots of PC1 and PC2, (**B**) top metabolite contribution percentages to PC1 and PC2, and (**C**) the dendrogram cluster.

**Figure 3 molecules-26-01415-f003:**
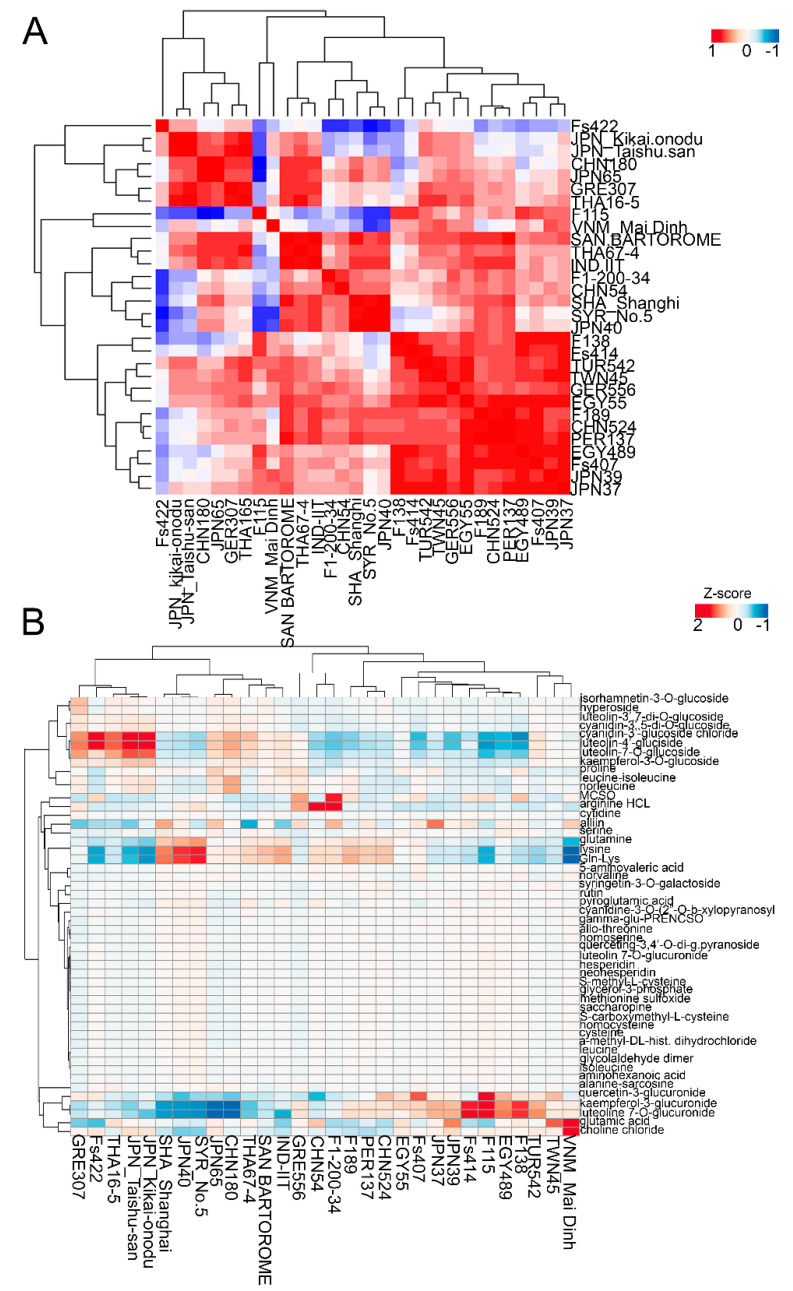
Correlation analysis and heatmap clustering of the 30 garlic accessions: (**A**) Pearson correlation analysis and (**B**) heatmap clustering of the normalized metabolite matrix.

**Table 1 molecules-26-01415-t001:** Garlic accessions used in the present study.

World Collection No.	Sample Name	Region of Origin
AsWC146	JPN37	Japan
AsWC147	JPN39	Japan
AsWC148	JPN40	Japan
AsWC149	TWN45	Taiwan
AsWC150	CHN54	China
AsWC151	EGY55	Egypt
AsWC154	JPN65	Japan
AsWC158	PER137	Peru
AsWC159	CHN180	China
AsWC162	GRE307	Greek
AsWC169	EGY489	Egypt
AsWC173	CHN524	China
AsWC175	TUR542	Turkey
AsWC177	GER556	German
AsWC181	F115	Central Asia
AsWC183	F138	Central Asia
AsWC186	F189	Central Asia
AsWC189	F1-200-34	Central Asia
AsWC193	Fs407	Central Asia
AsWC195	Fs414	Central Asia
AsWC196	Fs422	Central Asia
AsWC198	SAN BARTOROME (Gatur)	Turkey
AsWC201	SYR_No.5	Syria
AsWC202	IND-IIT	India
AsWC205	VNM_Mai Dinh	Vietnam
AsWC208	THA67-4	Thailand
AsWC209	THA16-5	Thailand
AsWC214	JPN_Taishu-san	Japan
AsWC224	SHA_Shanghi	China
AsWC226	JPN_Kikai-onodu	Japan

## Data Availability

The data presented in this study are available in Appendix A.

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
