# Peer review of "Comprehensive Metabolite Profiling in Genetic Resources of Garlic (Allium sativum L.) Collected from Different Geographical Regions"

_molecules, 2021, doi:10.3390/molecules26051415_

Round 1

Reviewer 1 Report

The manuscript is an interesting overview of the relation among the garlic genotypes, the origins, and the distinctive nutritive/nutritional traits. Thus the work should fit well within the scope of the journal.

The experiments are well described and the statistical analysis is robust. The standard of English is good. In general, the paper needs a few improvements, a little effort now aiming for great clarity.

  1. The Results section resulted well described along with the figs./images. The discussion section is too broad, but it includes very often the description of many results. In addition, the conclusions (clearly reported in the abstract, well supported by data) seems missing in the discussion section. I suggest merging the Sections (Results+Discussion+Conclusions), if the journal guideline allows it, thus to be clear, precise, and concise. By describing the results. you can parallelly discuss them, and draw conclusions.
  2. Be careful to digit correctly the measurement units.

Author Response

Dear Editor,

Thank you very much for handling the review process of our manuscript entitled “Comprehensive Metabolite Profiling in Genetic Resources of Garlic (Allium sativum L.) Collected from Different Geographical Regions” (Manuscript ID: molecules-1082126)

We have carefully addressed all the comments and suggestions raised by you and the Reviewers in the “Responses to the Reviewers’ Comments”, and revised the manuscript accordingly. All the changes made in the revised manuscript in response to the Reviewers’ comments were highlighted in red.

We hope that our revised manuscript is now acceptable for publication in Molecules

We look forward to receiving your favorable decision soon.

Sincerely,

Masayoshi Shigyo

Yamaguchi University

[email protected]

Tel.: +81-839-335-842

#Reviewer 1

The manuscript is an interesting overview of the relation among the garlic genotypes, the origins, and the distinctive nutritive/nutritional traits. Thus the work should fit well within the scope of the journal.

The experiments are well described and the statistical analysis is robust. The standard of English is good. In general, the paper needs a few improvements, a little effort now aiming for great clarity.                                                                                                                                  Response: We are very happy that the Reviewer positively evaluated our manuscript. We have revised the manuscript carefully, taking into account all the constructive comments and suggestions made by the Reviewers, which have helped us greatly improve the quality of our manuscript.

Q1.The Results section resulted well described along with the figs./images. The discussion section is too broad, but it includes very often the description of many results. In addition, the conclusions (clearly reported in the abstract, well supported by data) seems missing in the discussion section. I suggest merging the Sections (Results+Discussion+Conclusions), if the journal guideline allows it, thus to be clear, precise, and concise. By describing the results. you can parallelly discuss them, and draw conclusions.

Response: We appreciate the Reviewer for his/her critical and constructive comments. Based on Journal style and format the ‘Result’ and ‘Discussion’ sections should be separated. Following the Reviewer advice, we added and discussed several research articles related to comparative metabolites in garlic varieties in the ‘Discussion’ section in the revised manuscript.

Q2. Be careful to digit correctly the measurement units.

Response: We appreciate the Reviewer for his/her critical and constructive comments. Following the Reviewer advice, we revised and checked all the digit and measured units in the whole manuscript.

Reviewer 2 Report

In this manuscript, the authors established comprehensive metabolite profiling in genetic resources of garlic collected worldwide. The study investigated the metabolic profiles in the leaf tissue of 30 garlic accessions collected from different geographical regions using LC/MS techniques. The manuscript is well written and presented. However, the main concern I have is related to sample collection. The aims of this study is to investigate the metabolite profiling in genetic resources from different geographical regions.

Sample collection, mainly from the Asia region, and only one from Peru, Greek, Egypt, Turkey and Germany are not enough population to represent for the region. Therefore, the result of this study about the metabolite profiling in genetic resources from different geographical regions is not reliable.

Samples were collected from 1970s, how authors/the university maintain them until 2012?   

Comments:

L60 rewrite

L254: -high anthocyanin-related metabolites including cyanidin and luteolin glucosides as should be high anthocyanin-related metabolites including cyanidin-based and luteolin-based glucosides

Materials and methods: Leaf collection is a very important step. Leaf age, leaf position, leaf size???

Supplementary Materials: Tables missing units for each measurement. Is it mg/100g DW or FW?

Figure 2A and figure 3 have a very low resolution; compound codes: X….. are too long and hard to follow. Suggest change the current compound codes, cluster II of figure 2A (left) is too messy to see, need to improve.

L174 accumulations of cyanidin-based glucoside and Luteolin-based glucoside

Author Response

#Reviewer 2

Q3. In this manuscript, the authors established comprehensive metabolite profiling in genetic resources of garlic collected worldwide. The study investigated the metabolic profiles in the leaf tissue of 30 garlic accessions collected from different geographical regions using LC/MS techniques. The manuscript is well written and presented. However, the main concern I have is related to sample collection. The aims of this study is to investigate the metabolite profiling in genetic resources from different geographical regions.

Sample collection, mainly from the Asia region, and only one from Peru, Greek, Egypt, Turkey and Germany are not enough population to represent for the region. Therefore, the result of this study about the metabolite profiling in genetic resources from different geographical regions is not reliable.

Response: We appreciate the Reviewer for his/her critical and constructive comments. In our garlic collections we have more than 107 garlic accessions, but due to the high cost for metabolome analysis of all these accessions we selected a represented accession for each country, with several accessions from each region with special focus on Asian region. Following the reviewer’s advice, we revised the title into

“Comprehensive Metabolite Profiling in Genetic Resources of Garlic (Allium sativum L.) Collected from Different Geographical Regions”

Additionally, we revised the sample collection information in the abstract to highlight that many accessions were collected from Asian region as below:

“…collected from different geographical regions, with special focus on Asian region” L29

Q4. Samples were collected from 1970s, how authors/the university maintain them until 2012? 

 Response: We appreciate the Reviewer for his/her constructive comment. Following the Reviewer’s advice, the information regarding the maintenance of garlic accessions was revised in the ‘Material and Methods” section as below:

“To maintain these important garlic genetic resources, ten cloves from three to five bulbs of each accession were vegetatively propagated every year in October and harvested by end of Jun. Then entire garlic plant (bulb, roots, and stalk) were hanged and kept in dry place with good air ventilation inside the Greenhouse.” L338-L341

Comments:

Q5. L60 rewrite

Response: We appreciate the Reviewer for his/her suggestion. Following the Reviewer’s advice, we revised the sentence as below:

“For example, the high temperature during thermal processing decreased amino acid, allicin and moisture contents of black garlic, whereas an increase in the total phenolic, reducing sugar, and organic acid contents was observed, indicating that temperature had a significant influence on the quality and flavor of black garlic [6].” L60-61

Q6. L254: -high anthocyanin-related metabolites including cyanidin and luteolin glucosides as should be high anthocyanin-related metabolites including cyanidin-based and luteolin-based glucosides

Response: We appreciate the Reviewer for his/her suggestion. Following the Reviewer’s advice, we revised the sentence as below:

“…metabolites including cyaniding-based and luteolin-based glucosides…” L295

Q7. Materials and methods: Leaf collection is a very important step. Leaf age, leaf position, leaf size???

Response: We appreciate the Reviewer for his/her constructive comment. Following Reviewer’s advice, the information related to leaf age, position and size were revised in the “Material and Method” section as below:

“…Two month later after planting, the youngest leaves, which over four cm in length, were sampled” L364-365

Q8. Supplementary Materials: Tables missing units for each measurement. Is it mg/100g DW or FW?

Response: We appreciate the Reviewer for his/her constructive comment. We checked and confirm the units in Table S1, whereas the data in Table S2 is relative intensities using LC/MS not absolute quantification. The unit is g inulin equivalent/100g FW for measurement of fructan and mg/g DW for saponin.

Q9. Figure 2A and figure 3 have a very low resolution; compound codes: X….. are too long and hard to follow. Suggest change the current compound codes, cluster II of figure 2A (left) is too messy to see, need to improve.

Response: We appreciate the Reviewer for his/her constructive comments and suggestions. Following the Reviewer’s advice we increased the resolution of the figures 2 and 3 up to 700 dpi, in addition the letter size increased to improve the readability. Regarding Fig.2A left panel, we also improve the size, however as Reviewer know that this is score plot, which includes more than 100 metabolites, thus the overlapping of metabolite codes will occur regardless to the increase or decrease of the size and style. In addition, original images with high resolutions were uploaded into the Journal system.

L174 accumulations of cyanidin-based glucoside and Luteolin-based glucoside

Response: We appreciate the Reviewer for his/her suggestion. Following the Reviewer’s advice, we revised the sentence as below:

“….accumulations of cyaniding-based and luteoline-based glucosides” L179

Reviewer 3 Report

Comments:

In the title, the author stated that metabolite profiling in genetic resources of garlic has been collected worldwide, but from Europe only samples from Germany are included, as we know that Spain had the strongest breeding program for vegetables of the genus Allium and also from America only one sample from Peru is included? Please redesign the title that would match the content of the article!

In the Materials and Methods section:

Please add the name of the fertilizer and the manufacturer (line 298-299);

Please add the detailed description of the chemical analysis of the soil where the garlic plants were cultivated!

Why did the authors apply the nitrogen as amonium sulfate rather than nitrate? Is there a problem with the pH of the soil? Explain.

Regarding the fact that the experiments were conducted outdoors (in the open field), please add the data of climatic conditions during the experimental period!

In the "Discussion" section, please comment on changes in the metabolite profile that could be a consequence of the growing conditions

Author Response

#Reviewer3

Q10. In the title, the author stated that metabolite profiling in genetic resources of garlic has been collected worldwide, but from Europe only samples from Germany are included, as we know that Spain had the strongest breeding program for vegetables of the genus Allium and also from America only one sample from Peru is included? Please redesign the title that would match the content of the article!

Response: We appreciate the Reviewer for his/her constructive comments. Following the Reviewer’s advice, we revised the title as below:

Comprehensive Metabolite Profiling in Genetic Resources of Garlic (Allium sativum L.) Collected from Different Geographical Regions

In the Materials and Methods section:

Q11. Please add the name of the fertilizer and the manufacturer (line 298-299);

Response: We appreciate the Reviewer for his/her constructive comments. Following the Reviewer’s advice, the name of fertilizers and manufacture were included in the Material and method section as below:

“CDU-S555 (JCAM AGRI. Co., Ltd., Tokyo, Japan), Ecolong 413-100 (JCAM AGRI. Co., Ltd., Tokyo, Japan), and Ecolong 413-140 (JCAM AGRI. Co., Ltd., Tokyo, Japan) were applied at 130, 360 and 130 kg/ha, respectively before planting” L360-363

Q12. Please add the detailed description of the chemical analysis of the soil where the garlic plants were cultivated!                                                                                                         Response: We appreciate the Reviewer for his/her constructive comments. We did not carry out soil chemical analysis in this experiment.

Q13. Why did the authors apply the nitrogen as amonium sulfate rather than nitrate? Is there a problem with the pH of the soil? Explain.                                                                          Response: We appreciate the Reviewer for his/her constructive comments.  Ammonium sulfate was added as nitrogen source and to adjust the pH of the soil between 6 and 6.5. The fertilizer information was revised in the text as below:

“Then, a compound fertilizer was applied before planting, and 100 kg/ha ammonium sulfate for N source and adjust soil pH (pH 6.0-6.5), 100 kg/ha potassium chloride as K source and 120 kg/ha calcium…” L343-345

Q14. Regarding the fact that the experiments were conducted outdoors (in the open field), please add the data of climatic conditions during the experimental period!

Response: We appreciate the Reviewer for his/her constructive comments.  The climate condition during cultivation was added in the ‘Material and Method’ section as below:

“Monthly mean temperature (°C), precipitation (mm) and accumulation of daylength (h) (minimum / maximum) were 16.4 °C (4.7 °C / 29.2 °C), 191.6 mm (17.0 mm / 648.0 mm) and 194.3 h (95.0 h / 279.2 h) in 2017-2018, 16.8°C (6.0 °C /28.6 °C), 108.6 mm (22.0 mm / 342.5 mm) and 174.4 h (103.3 h/ 262.1 h) in 2018-2019, respectively (data from Japan Meteorological Agency)” L367-371

Q15. In the "Discussion" section, please comment on changes in the metabolite profile that could be a consequence of the growing conditions

Response: We appreciate the Reviewer for his/her constructive comments. . Following the Reviewer’s advice, we added and discussed several research articles related to comparative metabolites in garlic varieties and growth conditions in the ‘Discussion’ section.

Round 2

Reviewer 2 Report

Line 180: cyanidin-based not "cyaniding-based".